# Towards the Latent Transcriptome

## Abstract

In this work we propose a method to compute continuous embeddings for kmers from raw RNA-seq data, in a reference-free fashion. We report that our model captures information of both DNA sequence similarity as well as DNA sequence abundance in the embedding latent space. We confirm the quality of these vectors by comparing them to known gene sub-structures and report that the latent space recovers exon information from raw RNA-Seq data from acute myeloid leukemia patients. Furthermore we show that this latent space allows the detection of genomic abnormalities such as translocations as well as patient-specific mutations, making this representation space both useful for visualization as well as analysis.

## 1 Introduction

The fundamental issue we would like to address in this paper is the need for a flexible representation of RNA-Seq experimental data. Standard RNA-Seq analysis pipelines discard rich information for a more canonical result (Valbuena et al., 1978; Zielezinski et al., 2017). This information may be crucial, since diseases such as cancer are known for their high mutational burden, multiple rearrangements and unconventional genomes, which do not fit the assumptions of the standard RNA-Seq pipeline, that uses hand-crafted features as a basis for analysis. To address this, we learn a latent space that captures gene-like structures from raw RNA-Seq data. We find that our proposed model successfully recovers information on gene sequence similarity, mutations, and chromosomal rearrangements.

The transcriptome is a subset of all possible sequences of the genome that are used by the cell at any given moment and constitutes less than 2% of all genomic sequences (if we consider only one cell type). Of this transcriptome, only a small amount is captured in standard RNA-Seq analysis pipelines, mainly transcripts that encode proteins (total of 20-60k sequences). The goal of these pipelines is to count the relative abundance of each transcript in the cell.

The raw data actually contains much more information than just gene abundance, namely patient-specific mutations and chromosomal rearrangements. RNA-Seq experiments yield very rich data, that can be informative both in terms of sequence abundance as well as sequence composition. Reducing this rich data to only the detection of annotated genes (which are "hand-crafted" features of the sequence) is not appropriate for analysis. Indeed, although the simple story relating each gene to a protein is correct to first approximation, there are important phenomena such as gene homology, patient-specific mutations, translocations and other genomic alterations that are excluded from the analysis, despite their presence in the data.

In this work, we consider the problem of including the rich patient-specific sequence information from RNA-Seq data via a continuous representation that will account for both gene expression as well as mutations and chromosomal rearrangements. We propose a model which learns gene-like representations from the raw patient-specific sequence RNA-Seq data. We study how this model handles situations that are standard in cancer genomics but considered edge case in standard pipelines.

## 2 RELATED WORK

### 2.1 THE STANDARD RNA-SEQ ANALYSIS PIPELINE

Once RNA is extracted from cells in the lab, it is processed by a sequencer. Individual RNA sequences are fragmented into short 100-200 base pair sequences (each of which is called a *read*) before entering the sequencer and then processed in bulk. A sequencing experiment produces a vast amount (billions) of short character sequences (reads, $R$), each character (A,C,T,G) representing each of the four nucleic acids (Figure 1A). A good analogy of the way RNA-Seq experiments are done would be to compare the output of a sequencer to the output of a shredder. To deal with a shredder-generated output, a reference text would be helpful. This reference text (approximately like the text of the shredded document) is used to look-up the shredded sequences to determine their regions of origin in the text. This works well so long as the true underlying text and the reference are fairly close and that their differences are local. Unfortunately, this assumption breaks in the case of cancer-induced mutations.

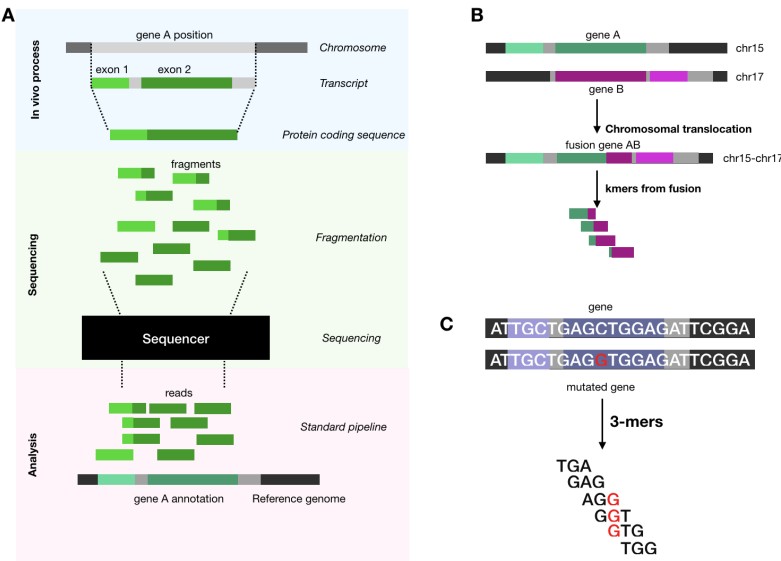

Figure 1: A) Standard RNA-Seq analysis pipeline, including alignment of sequencing reads. B)Translocation between two chromosomes and creation of a fusion gene AB. Kmers overlapping the fusion will partially match both original sequences. C) Every patient-specific mutation creates at least *k* new kmers. Here new kmers are shown for k=3

A reference genome exists for most of the widely studied species, see Figure 1 for the standard processing pipeline. In this reference genome, gene locations and exact sequence are annotated according to the current revision. The reference genome is renewed every five years or so, to take into account new discoveries in genomics. Zielezinski and colleagues report that sequence alignment methods fail in the edge cases of high sequence similarity (homology) (Zielezinski et al., 2017). Moreover, multiple sequence alignment is an NP-hard problem (Chatzou et al., 2016), that is sped up using heuristics and yielding the "most probable" alignment, which adds a level of uncertainty to results.

In the event of a chromosomal translocation (Figure 1B), a common occurrence in cancer, reads that cover both parts will only be matched to the un-fused sequences based on a sequence similarity score (typically $< 50\%$ match) (Zielezinski et al., 2017). This makes chromosomal rearrangements such as translocations undetectable by RNA-Seq, since they are not part of the reference genome. For this specific case, we deem the reference genome inappropriate. Indeed, reference based methods yield a relative abundance measurement of genes, which are by definition, hand crafted features.

## 2.2 Merging RNA-Seq experiments with additional genomic data

Cancer cells have often unconventional genomes, many showing chromosomal rearrangements, mutations, copy-number variations (CNV) and repeated regions(Weinstein et al., 2013). The correct identification of these rearrangements is a non trivial challenge for reference genome-based pipelines, since these modifications are not included in the reference genome.

Many recent advances in the field of cancer research have become possible either by combining standard pipeline-derived RNA-Seq data with other sequence-specific data, such as SNP arrays, miRNA-Seq and even whole genome sequencing (Koboldt et al., 2012; Gerstung et al., 2015; Hu et al., 2016; Gao et al., 2013; Weinstein et al., 2013; Liu et al., 2017). While combining data does yield good results, genomic variant analysis almost always requires a predefined sequence of interest to orient the search.

Other teams preferred to develop reference-free methods for variant calling. One such reference-free method, *km*, stores n-grams (also known as kmers in computational biology) coming directly from reads into a De Bruijn graph-like structure (Audemard et al., 2018). They argue that since only a small part of the genome is expressed, variant detection can be limited to the transcriptome. Their tool, *km*, uses only the abundance of these kmers in patients to detect genomic abnormalities. While this method does not depend on any type of reference genome, it still shares the same problem as the ones that combine RNA-Seq with other experiments; all these methods require a predefined sequence for analysis. In other words, to find an abnormality in the cancer samples, one must know in advance the exact abnormal sequence to look for.

## 2.3 Representation learning for biological sequences

After the success of distributed representations in NLP (Mikolov et al., 2013) some teams have attempted to create distributed representations for biological sequences. Asgari & Mofrad (2015) adapted Word2Vec to create BioVec, distributed representations for biological sequences, based solely on sequence similarity. They report that their representation captures biochemical properties of proteins such as mass, volume and charge. Jaeger et al. (2018) has also extended the model to chemical compounds and observe that modeling chemical compounds with vectors yields a better performance in bioactivity prediction tasks. This work focuses on a different aspect of the problem. We consider the idea of using an unsupervised learning approach to directly learn a representation for RNA-Seq data from scratch, without the need for a reference genome and the corresponding definition of genes as clearly delineated and non-overlapping regions in the overall sequence.

## 3 Method

We represent the raw data as $\mathcal{R}$, where each read $\mathbf{r} \in \mathcal{R}$ is a sequence of length 100, where $r_j \in \{A, C, G, T\}$. We define a kmer (ngram) $\mathbf{x}$ of length $k$ as a subsequence for some read $\mathbf{r}$ from positions $l$ to $l + k - 1$:

$$\mathbf{x} = \mathbf{r}_{l:l+k-1}$$

Each patient $i$ generates a set $\mathcal{R}_i$ of reads. We extract kmers of length 24 from all the reads $\mathcal{R}_i$ to generate the kmer table $\mathbf{X}_i$, one for each patient. Table $\mathbf{X}_i$ contain $K_i$ unique kmers from reads $\mathcal{R}_i$.

$$\mathbf{X}_i = \{\mathbf{x}_1, \mathbf{x}_2, ..., \mathbf{x}_{K_i}\}$$

This table as well as the patient's id are used by our model to learn the kmer representation space.

The method uses the factorized embedding model (Trofimov et al., 2017) combined with an RNN. This specific method was previously shown to provide a multi-view embedding of genomics data. We however found that it lacked patient-specific information on the RNA sequence level. Our modification of this model allows for the integration of this sequence information to the factorized embeddings model. Concretely, the model is given a pair of inputs $< \mathbf{x}_{ij}, \mathbb{1}_i >$ and predicts the corresponding count $y_{ij}$. We represent each nucleotide as a $one - hot$ vector. For example, adenine and cytidine, $A$ and $C$, would be represented by respectively $[0, 0, 0, 1]$ and $[0, 0, 1, 0]$.

For each input pair, a corresponding pair of embeddings $< \mathbf{u}_{ij}, \mathbf{v}_i >$ is then computed. For $\mathbf{v}_i$, a lookup table is used, so each patient has its own embedding coordinates in patient embedding

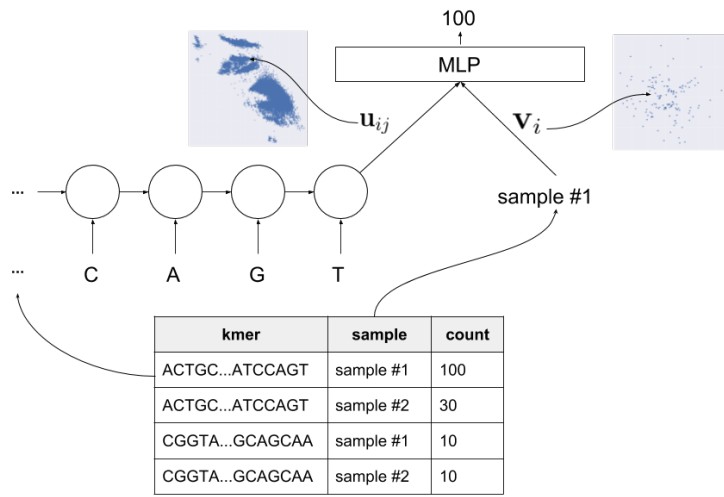

Figure 2: The model, where we predict the count of each kmer based on the their sequence (Using an RNN) and the sample id. We can then visualise the learned embeddings, or use them for some downtream tasks.

space. For $\mathbf{u}_{ij}$, we use a bidirectional LSTM to compute the embedding coordinates (Hochreiter & Schmidhuber, 1997), where the equations are given by:

$$f_t = \sigma(W_f x_t + U_f h_{t-1} + b_f) \tag{1}$$
$$i_t = \sigma(W_i x_t + U_i h_{t-1} + b_i) \tag{2}$$
$$o_t = \sigma(W_o x_t + U_o h_{t-1} + b_o) \tag{3}$$
$$\tilde{c}_t = tanh(W_c x_t + U_c h_{t-1} + b_c) \tag{4}$$
$$c_t = f_t \odot c_{t-1} + i_t \odot \tilde{c}_t \tag{5}$$
$$h_t = o_t \odot tanh(c_t). \tag{6}$$

The embeddings $\mathbf{u}_{ij}$ are then computed as follow:

$$\mathbf{u}_{ij} = f_{linear}([\mathbf{h}_{rnn}^{\leftarrow}, \mathbf{c}_{rnn}^{\leftarrow}, \mathbf{h}_{rnn}^{\rightarrow}, \mathbf{c}_{rnn}^{\rightarrow}]). \tag{7}$$

Where $\mathbf{h}_{rnn}^{\leftarrow}$ and $\mathbf{c}_{rnn}^{\leftarrow}$ are the hidden states of the forward RNN, and $\mathbf{h}_{rnn}^{\rightarrow}$ and $\mathbf{c}_{rnn}^{\rightarrow}$ are the hidden states of the backward RNN. A linear function $f_{linear}$ is learned to reduce the dimensionality of the embedding $\mathbf{u}_{ij}$ for visualisation purposes.

The two embeddings are then fed to a MLP to predict the corresponding count:

$$\hat{y}_{ij} = f_{pred}([\mathbf{u}_{ij}, \mathbf{v}_i]). \tag{8}$$

We use the quadratic loss as a training objective:

$$\mathcal{L} = \sum_{i,j} (\hat{y}_{ij} - y_{ij})^2 \tag{9}$$

Intuitively, kmers that come from the same gene would have the same count, since gene expression is counted in terms of transcripts. The model is thus encouraged to group kmers in embedding space $U$ that generally occurs together across all patients. Similarly, patients that have the same kmer occurrence profile should be grouped together in the embedding space $V$. A plan of the model is presented in Figure 2.

## 4 EXPERIMENTS

We divide our experiments into three tasks. Each task aims to test the behaviour of our model when presented with DNA sequences that have specific properties. Using all reads from an organism

would be optimal for experiments however the scale of the computation is currently intractable (as discussed in the Limitations section). Instead we focus on just specific regions of the genome that contain only a few genes. We determine this region using the a standard alignment pipeline but include the entire region of the gene (not just exons) and also use the aligned data per patient which includes patient specific mutations.

- Task 1: Embedding of genes sequences of high similarity.
- Task 2: Embedding of genes with different sequences
- Task 3: Embedding of genes that participate in a chromosomal translocation.

## 4.1 DATA

For all our experiments we used aligned, unquantified RNA-Seq data (BAM format files) from The Cancer Genome Atlas (TCGA) (Weinstein et al., 2013). The dataset contains 149 patients with acute myeloid leukemia (AML), a cancer well known for its multiple chromosomal rearrangements. For all our experiments, we isolated the reads (Figure 1) that span specific regions of interest, in order to speed up computation (Table 1). Upon extraction of the reads from the BAM files, we obtained a total of 22,008,292 kmers for all patients, with an average of 125,000 kmers per patient. We excluded kmers with occurrence of $< 2$, as these are most likely sequencing errors. All kmer occurrences were log-normalized.

Table 1: Regions of interest isolated for all experiments in this paper

| gene name | chromosome | start-stop | manuscript section |
|:---:|:---:|:---:|:---:|
| ZFX | chrX | 24148934-24216255 | 4.3 |
| ZFY | chrY | 2934416-2982508 | 4.3 |
| PML | chr15 | 73994673-74047812 | 4.4 |
| RAR$\alpha$ | chr17 | 40309171-40357643 | 4.4 |

## 4.2 EXPERIMENTAL DETAILS

The Bidirectional LSTM model had 2 layers with 256 hidden units. The size of each embedding (i.e. the output of the bi-RNN and sample id) was of size 2 for visualisation purposes. The prediction model is a two layers MLP of size 150 and 100 units respectively with the ReLU activation function. Each model was trained for 200 epochs with RMSProp with a learning rate of 0.001, $\alpha = 0.99$ and a momentum of 0.

## 4.3 TASK 1: REPRESENTATION OF GENES WITH HIGHLY SIMILAR SEQUENCES

For the first task, we wanted to test how kmers from two highly similar sequences would be embedded. The human genome contains many highly similar sequences and it is generally believed that similar sequences can have a similar function in the cell (Pearson, 2013). Although this is not always the case, we chose two genes that share a significant amount of overlap in sequence: ZFY and ZFX genes. It has been reported that they encode proteins of almost identical structure, containing 13 zinc fingers (Palmer et al., 1990; Schneider-Gädicke et al., 1989). While these genes are similar in sequence, they are located on two different chromosomes in the genome. Moreover, ZFY gene is only present in male patients, since it is located on the Y chromosome (Palmer et al., 1990). We argue that this is a case where standard gene annotation separates these two genes into two features but does not take into account their homology.

We obtained the reference sequences from the reference genome and found that they share 206 kmers (Figure 3A) in the protein coding regions. We expected the model to group the common (all sexes) kmers in one region and possibly push aside in embedding space the kmers that were ZFY gene-exclusive.

While our model does not use a reference genome for representation, in order to identify the kmers in embedding space, we obtained the reference sequences (GRCh38/hg38 Assembly of December

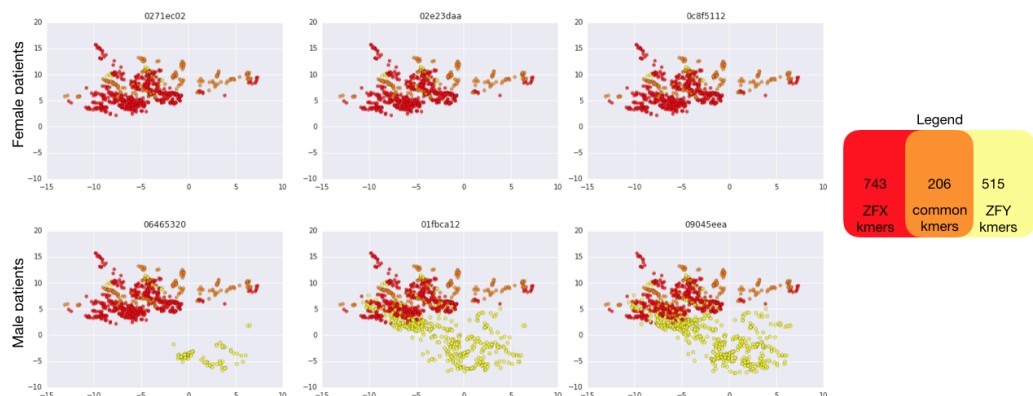

Figure 3: Embedding of homologous genes A) Embedding space for kmers in male and female patients. ZFY and ZFX genes share 206 kmers. Points are coloured according to their matching sequence of origin

2013) for both genes. We then coloured each kmer in embedding space according to the gene sequence it belongs to. We also annotated kmers that match both sequences. As expected, kmers from female patients were all grouped in the same region, while ZFY-gene exclusive kmers from male patients were located in another region of the embedding space (Figure 3A).

This result supports our claim that when genes share a large part of their sequence, a representation that groups these similar sequences together would contain more information. Although standard pipelines would label these genes as different features and quantify them separately, reads that would fall in the common region would be ambiguous (i.e. matching both regions) and therefore would be either clipped or redistributed according to the mapping software strategy (Kim et al., 2013). We argue that our representation captures both gene expression (via the kmer counts) as well as the gene sequence similarity.

## 4.4 TASK 2: REPRESENTATION OF GENES WITH DIFFERENT SEQUENCES

For our second task, we tested the embeddings model using two gene regions that are highly different (no homology). This task verifies how the model would arrange in embedding space kmers that come from different genes. Unlike the previous task, the two gene sequences are very different and the model has to learn sequence similarity between kmers as well as kmer abundance variations along the gene. This task is in line with the problems that come from scaling up our method to an entire organism.

The first gene of interest that we chose is the promyelocytic leukemia gene (PML), a tumor suppressor gene, involved in the regulation of p53 response to oncogenic signals. The second gene is the retinoic acid receptor alpha gene (RAR$\alpha$), a gene involved in many core cellular functions such as transcription of clock genes. These genes do not share a significant amount of kmers and serve as an example to test how the model will arrange kmers when the sequences are different.

By matching patient kmers to each of the exons from the reference genome (we used GRCh38/hg38 Assembly of December 2013) of the PML and RAR$\alpha$ genes, we notice that the embedding model grouped kmers in embedding space by matching exon (Figure 4A). Exons are subsequences of the transcripts that actually encode the protein (Figure 1A). The information recovered by the standard RNA-Seq analysis pipeline corresponds to the exons for these two genes, since they code for both proteins. We conclude that our model recovers correctly information obtainable by the standard RNA-Seq analysis pipeline, when looking at two protein coding regions.

Moreover we observe that there seems to be a discrepancy in the kmer counts within the gene transcripts. Indeed, for both genes, some exons seem to be present at higher counts (Figure 4B). Although the canonical gene expression model states that each transcript is copied in its entirety before being spliced and then translated to a protein (Figure 1A), there exists the highly documented

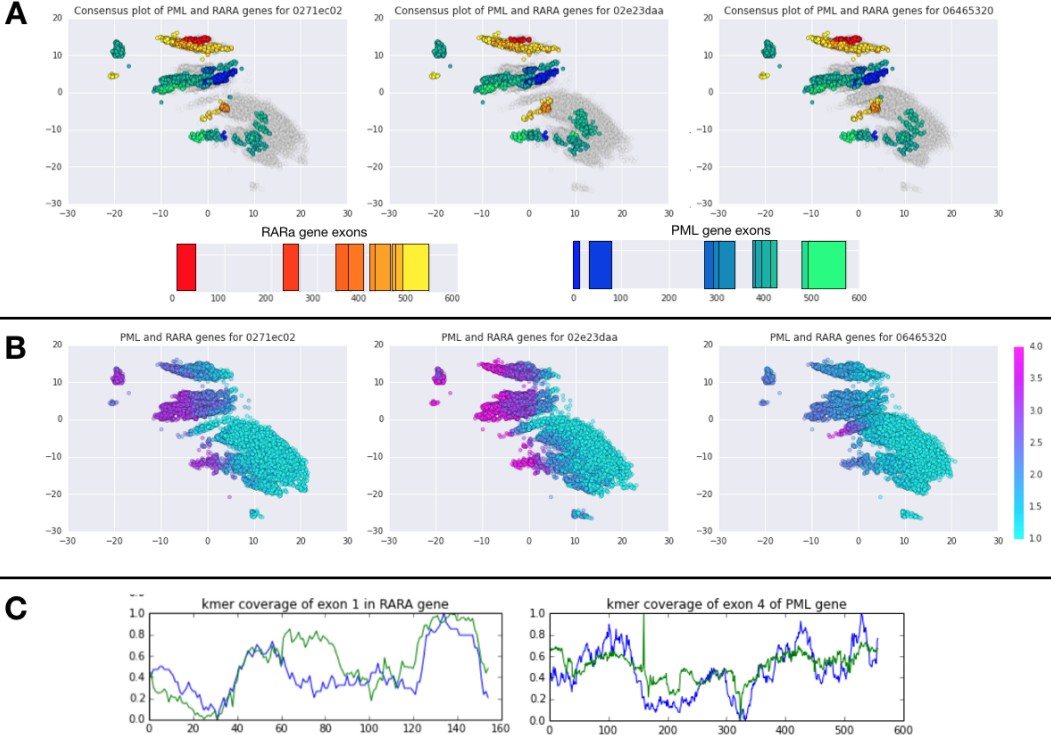

Figure 4: Kmer embedding space. A) Kmer embedding space is shown with kmers of 3 individuals. Points are coloured according to the corresponding kmer position in exons for both PML gene (blue), RARa gene (red) and unknown kmers in grey. B) Kmer embedding space is shown with kmers of 3 individuals. Points are coloured according to the kmer counts C) Actual kmer coverage (blue) of exons and prediction by the model (in green)

phenomenon of sequencing bias (Benjamini & Speed, 2012). This is an entirely experimental bias that is explained by the fact that RNA-Seq is done using an enzyme, polymerase. This enzyme has a bias in terms of G-C content of sequences.

The occurrence of guanine and cytosine (G and C) is measured by counting the G or C nucleotides and then dividing by the total length of the sequence. For example, the sequence *AATTGAGCGA* would have a G-C content of $(3G + 1C)/10 = 0.4$. We verified the relative exon composition and found that in general, kmers with a higher count overlap exons with a lower G-C occurrence (no figure shown).

Moreover, we compared the model's predicted kmer count and the actual count over various exons from both genes. We found that the predicted kmer count is proportional to the predicted kmer count, which confirms that our model captures fine-grained kmer abundance variations along exons (Figure 4C).

## 4.5 TASK 3: DETECTION OF ABNORMAL GENOMIC STRUCTURES

The PML and RAR$\alpha$ genes are also often involved in a chromosomal translocation (Figure 1B), which yields a new fusion gene the PML-RAR$\alpha$ in patients that have acute promyelocytic leukemia (de Thé et al., 1991; Lavau & Dejean, 1994). In our dataset, 10% of the patients are of the acute promyelocytic leukemia subtype and the choice of these two genes serves the dual role for testing the embedding model on two different genes as well as detecting a chromosomic translocation and resulting fusion gene.

Upon further examination of the embedding space for kmers in task 2, we noticed that a large amount of the kmers from both genes were not included in the exonic sequences according to the annotation

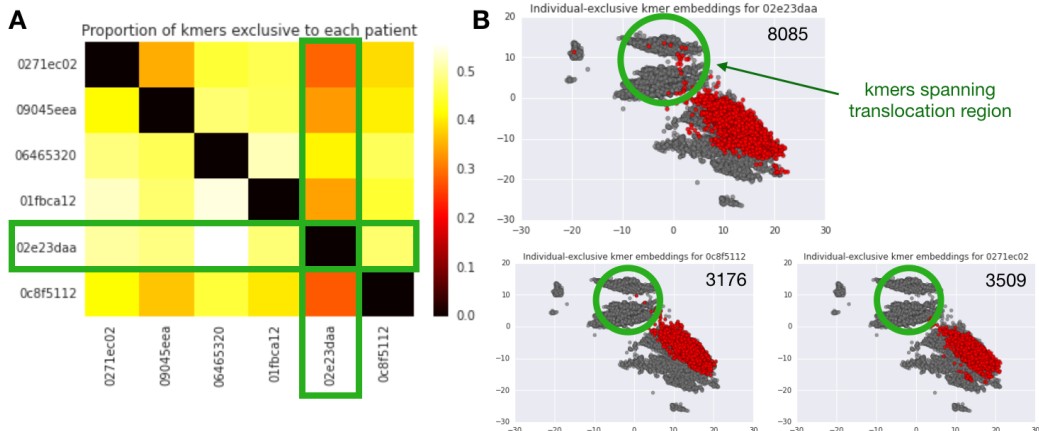

Figure 5: Individual-exclusive kmers. A) Heatmap showing the proportion of individual-exclusive kmers for each pairwise comparisons. B) Patient-exclusive kmers are highlighted in red. Green circle highlights kmers that span the translocation.

(Figure 4B). These sequences are excluded for any of the following reasons: i) the kmer sequence does not match (exact match) the reference sequence (example in figure 1C) ii) the kmer sequence is not included in the exonic region (3' and 5'-UTR and introns) or iii) the kmer matches to the t15:17 translocation site in the given patient (example in figure 1B). To better address this observation, we performed pairwise comparisons of patient kmer sets and found that patient 02e23daa has the most different transcriptome, with at most 50% kmers difference (Figure 5A).

We then compared patient-exclusive kmer sets (Figure 5B in red) and found that most patients have between 3-10k exclusive kmers. We isolated the kmers from patient 02e23daa and reassembled them into a consensus sequence. We used the software BLAST to perform multiple alignment of this sequence in the reference genome. We report that half of this sequence matches to the PML gene and the other half to the RAR$\alpha$ gene, a scenario matching that of a fusion gene, result of a chromosomal translocation (Figure 1B). This was confirmed by verifying the clinical data for patient 02e23daa, where the translocation was previously detected and annotated in the clinic (figure not shown). From these results, we conclude that our model captures real genomic abnormalities and allows to extract directly from the kmer embedding space the abnormal sequence.

## 5 LIMITATIONS

The main limitation of our model is its scalability. In all the tasks performed in this paper, we heavily restrained the number of kmers in the dataset. Indeed, without pre-filtering the BAM files, each sample would generate approximately 10-30 billion kmers (compared to 125,000 per sample in the current dataset). While this number is very high, we suggest that since kmers are overlapping by definition, it would be possible to sample the kmers while training and therefore greatly reduce the number of kmers seen by the model, thus reducing the processing time. Finally, we have not yet explored parallelizing the training onto multiple GPUs, which would greatly reduce the training time.

Finally, while we used the pre-aligned BAM file to filter our regions of interest, our goal is to optimize this model to move away from reference genomes entirely. To do so, we plan on using only two "seed" kmers and using the total kmer table, as a means to extract the kmers of interest. Indeed, exploring all paths supported by kmers between the two seeds is an (almost) reference-free method to generate kmer tables, without relying on the alignment product.

## 6 CONCLUSION

In this work we propose a model which learns gene-like representations from the raw RNA-Seq data. We show that this approach does not rely on the reference genome used in standard pipelines and instead learns a common representation by looking at multiple patients. We study how this model handles situations in cancer which are problematic for the standard RNA-Seq pipeline, such as homologous genes and translocations.

We believe that this model could extract genotype information from data, which is difficult to observe using standard alignment-based pipelines. This extracted information may provide a foundation for future methods to build on and better understand the underlying biology.

## ACKNOWLEDGEMENTS

This work is partially funded by the Canadian Institute for Health Research (CIHR), a grant from the U.S. National Science Foundation Graduate Research Fellowship Program (grant number: DGE-1356104), and the Institut de valorisation des donnees (IVADO). This work utilized the supercomputing facilities managed by the Montreal Institute for Learning Algorithms, NSERC, Compute Canada, and Calcul Quebec.

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

## 7    APPENDIX I

We generated a synthetic dataset of kmer counts. We created 20 strings of randomly selected $A, T, C, G$ characters that we termed "genes". We then sampled 5 individual count values for the genes from a Poisson distribution (Figure 6 C).

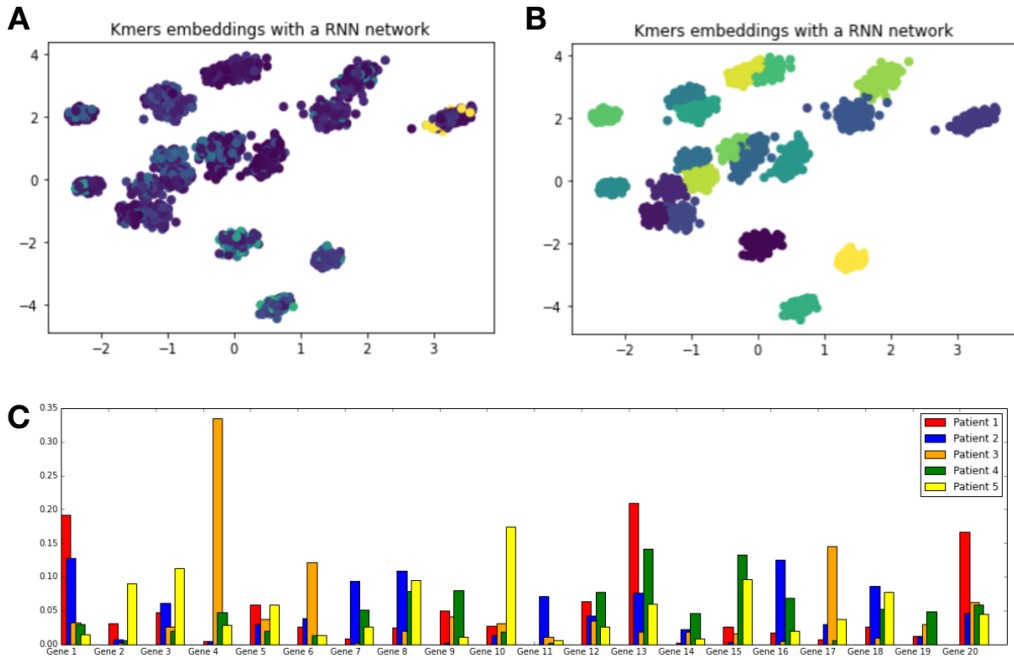

Figure 6: Synthetic dataset embedding experiment A) Kmer embedding space is shown with kmers of 5 individuals. Points are coloured according to the corresponding kmer count value B) Kmer embedding space is shown with kmers of 5 individuals. Points are coloured according to the corresponding gene C) Synthetic dataset kmer counts for 5 individuals

We trained the latent transcriptome model using this synthetic dataset and found that kmers grouped first by kmer similarity of sequence (Figure 6B). We also found that kmers grouped by kmer counts in specific patients (Figure 7 shows a couple examples). From here we conclude that both kmer sequence and kmer count are taken into account by the model to build embedding coordinates for kmers.

More investigation is needed to test the relative importance the sequence has over the kmer count, however, from this synthetic dataset experiment we conclude that kmers are first and foremost matched by sequence and then by similar kmer count profile across samples.

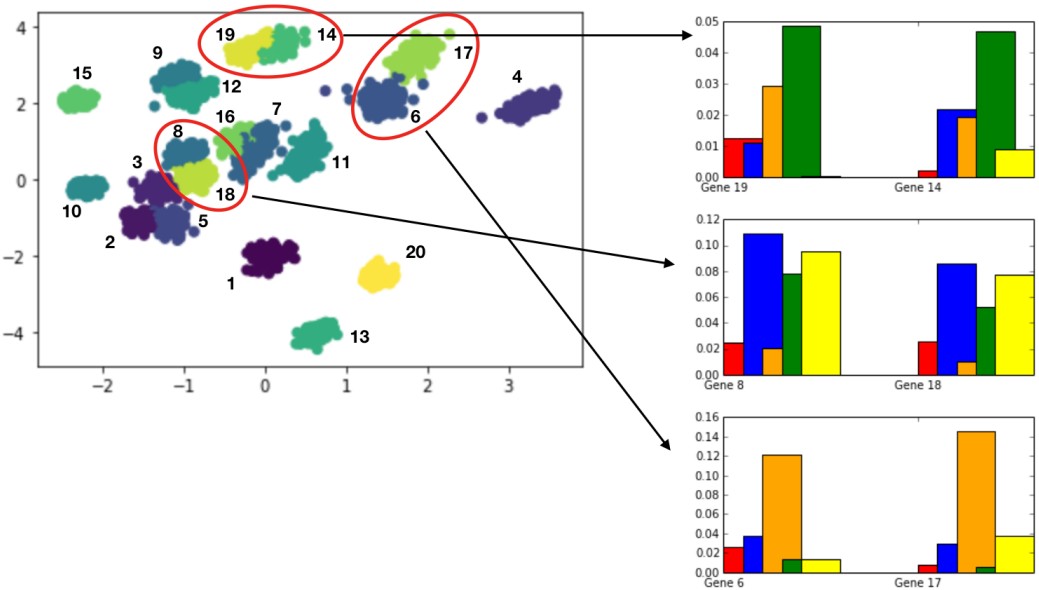

Figure 7: Synthetic dataset embedding experiment A) Kmer embedding space is shown with kmers of 5 individuals. Points are coloured according to the corresponding kmer count value B) Kmer embedding space is shown with kmers of 5 individuals. Points are coloured according to the corresponding gene C) Synthetic dataset kmer counts for 5 individuals

