# OpenReview forum: "Towards the Latent Transcriptome"
_ICLR.cc/2019/Conference_

### Official Review · AnonReviewer1 · 2018-10-31
**Interesting and potentially useful idea with impact, but some design choices need careful thought before it is useful in practice.**

**Rating:** 5
**Confidence:** 4

**Review:**


Summary:
The paper proposes an unsupervised method to learn a vector representation for short genomic sequences, so-called kmers (like n-grams in natural language processing). The method learns a representation that will result in a good predictor of kmer counts from the kmer sequence itself. The idea is that neighbouring kmers (from the same gene) would have similar counts (same gene expression), and hence would be embedded near to each other. The paper shows some small empirical experiments for 3 tasks: showing similar genes are close, able to distinguish different genes, and able to detect genomic structural variation.

This is an interesting idea, and would have large impact if done well. However the current approach has multiple weaknesses which leave the proposal less strong that it could be. The paper is written clearly, and while the idea is motivated from word2vec and derivatives, the application to kmers is original.


Overall comments:
- There are two issues conflated in the word scalability:
  1. computational scalability, where the authors need to run the method on a more realistic dataset and show that the LSTM converges.
  2. statistical scalability, which I will expand in the next point.
- The design of finding an embedding that will identify the count given a kmer has several weaknesses, which the paper did not address:
  1. Two genes could have similar expression, hence similar kmer counts, but different kmers.
  2. A kmer can appear in multiple genes, and the total count is the sum of all of them.
  3. Copy number variation (since the paper is interested in cancer) would affect counts
  4. Two kmers with only one or two differences could be due to SNPs. Should they be near or far?
  5. Should we learn a representation for each individual, or a representation for the population? Depending on how the sample id (and hence vector v_i) is used, one can get different effects.
- It seems wasteful that there is no representation learning for each individual, but instead it is just a fixed (arbitrary?) vector in a look up table.
- The choice of embedding dimension 2 seems to be driven by the fact that the authors wanted to visualize. This is tied up with a weakness that the paper does not measure the quality of the embedding, e.g. using reconstruction error. A good approach is to show that the resulting embedding is useful for some other prediction task (which is usually the reason we want to find an embedding). Reporting mean squared error for Figure 4C would also be helpful.

Minor typos/issues:
- page 3, Section 3: does j range over k-mers in x_{ij}? You also use it r_j in the first sentence.
- page 3, Section 3: read length = 100. kmer length = 24. This should be put in the experimental section. Furthermore due to reverse complements, it would be better to have an odd number for k, e.g. 23.
- page 3,4: using angle brackets to mean a pair is uncommon. Suggest tuple (u,v).
- page 4: The notation U in the description of the LSTM can be confused with two other things:
U is the kmer embedding space, and u_{ij} is the embedding vector.
- page 6: In the text you refer to Figure 3A, I assume you mean Figure 3.
- page 6, Figure 3: Unclear what the three columns are. I assume similar to Figure 4, they are three individuals.
- page 6, task 2: It is unclear how the reader can see that the information recovered by kmer2vec is the same information recovered by standard RNA-Seq analysis.
- page 8, first word: Not sure how Figure 4B shows what the sentence is trying to say.

---

> ### Author Response · Authors · 2018-11-27
> **Response to Reviewer 1**
>
> > This is an interesting idea, and would have large impact if done well. However the current approach has multiple weaknesses which leave the proposal less strong that it could be. The paper is written clearly, and while the idea is motivated from word2vec and derivatives, the application to kmers is original.
>
> We would like to thank Anonymous Reviewer 1 for the constructive comments and general ideas, we are very grateful for your review.
> While we could not cover everything in the rebuttal period time frame, here are some ways we addressed your comments
>
> > - The design of finding an embedding that will identify the count given a kmer has several weaknesses, which the paper did not address:
>   1. Two genes could have similar expression, hence similar kmer counts, but different kmers.
>   2. A kmer can appear in multiple genes, and the total count is the sum of all of them.
>   (...)
>   4. Two kmers with only one or two differences could be due to SNPs. Should they be near or far?
>
> We have in the beginning done some experiments on synthetic data. We added the plots generated from the synthetic data experiments to the Appendix. In short, we have generated 20 random strings of “nucleotides” we termed “genes”. We then extracted kmers from these strings and assigned them a count using a Poisson distribution. See the Appendix for more details.
>
> > 3. Copy number variation (since the paper is interested in cancer) would affect counts
>
> You are absolutely right! However, we did not address this variability in this extended abstract, fault of space. We plan on investigating this in future work and have added to the text the following:
> “Cancer cells have often unconventional genomes, many showing chromosomal rearrangements, mutations and repeated regions(Weinstein et al., 2013). Copy number variations (CNV) are also quite common in cancer cells (Shlien et al. 2009). ”
>
> > 5. Should we learn a representation for each individual, or a representation for the population? Depending on how the sample id (and hence vector v_i) is used, one can get different effects.
>
> Excellent observation! We did not include patient embedding space because we did not have the space in this extended abstract.

---

### Official Review · AnonReviewer3 · 2018-11-03
**Review of Kmer2vec method finds both method and evaluations unsuitable for stated aim, also lack of domain knowledge.**

**Rating:** 2
**Confidence:** 5

**Review:**

The stated contribution of the paper is the development of a model to learn continuous representations of k-mers from RNA sequencing experiments in an annotation-free manner. The paper motivates this model by considering analysis challenges faced in cancer genomics. This introduction serves well to frame the paper towards addressing these challenges. In particular, the authors highlight challenges faced in recognizing and quantifying patient/tumor-specific RNASeq based expression estimates involving structural variants and indels, which have and continue to be a challenge for existing tools.

Despite this, we are not enthusiastic about the paper for the following reasons:
Narrow and incomplete view of commonly used modern RNA-seq tools/pipelines and their application/use in biomedical research.

The proposed computational method is computationally intractable and is unlikely to ever scale to the genome-wide context.

Described experiments are without context to the existing literature of tools designed to address the biological challenge and by construction are not annotation free.

We further describe these reasons in the following subsections. Overall, we do not believe that the described model/experiments demonstrate utility for either the specific problems in cancer genomics that motivate the paper or the biomedical research field in general.

Narrow and incomplete view of RNA-seq tools/pipelines/application:
------------------
“Reducing this rich data to only the detection of annotated genes [...] is not appropriate for analysis”. Modern RNA-seq pipelines also perform quantification and differential analysis at a minimum.
The description of the standard RNA-seq experiment is problematic:
Sequencing is of cDNA after reverse transcription, not RNA.
Ignores paired end reads (especially with longer insert sizes for fusion detection)
Shredder poor analogy given multiple distinct reads from same sequence, known biases in process
“[...] multiple sequence alignment is an NP-hard problem”. This is true but irrelevant.
Chromosomal translocations are indeed hard to detect by RNA-seq, but not impossible. There are strategies implemented in commonly used tools such as STAR, kallisto, and others to detect these and other structural variants. Individual reads do not have to themselves cover the sequence where translocation occurs, instead read pairs can imply that the insert contains a translocation -- in this case, sequence similarity is much higher. Regardless, cheaper orthogonal assays exist that can detect these events.
“The standard RNA-Seq analysis pipeline has a mean processing times of 28 core hours for mapping with software TopHat, followed by an additional 14 hours of quantification”. See “Please stop using TopHat” (https://twitter.com/lpachter/status/937055346987712512?lang=en) by one of the authors of TopHat and the senior author of the cited paper. Standard pipelines use newer aligners like STAR which are substantially faster.
“Reference based methods yield a relative abundance measurement of genes, which are by definition, hand crafted features”. Relating results back to genes is important to be able to connect sequencing results back to known biology. We see the fact that there is no obvious map from the proposed method back to genetic information as a weakness.

Proposed computational method computationally intractable and unlikely to scale to genome-wide context
------------------------
Plus:
Paper does acknowledge that scalability is a limitation.
Minus:
Lower bound of range of unique kmers per sample without pre-filtering is 10 billion; note that storing counts uncompressed as 32-bit integers corresponds to over 37GB per sample.
Experiments are for only four genes, two at a time with a 2-dimensional embedding. Unclear how patterns will hold when considering k-mers from more genes simultaneously or how embedding could scale. Model formulation suggests that k-mers from co-expressed genes will have similar embeddings, which could complicate visual inspection.
Abstract states that “learned representation both useful for visualization as well as analysis”. Unclear what is done with model/embeddings besides visualization -- non-visual analyses are performed with k-mer counts before embedding. Identification of abnormalities are described only by visual inspection, which is unlikely to scale as more k-mers are added and/or if the dimensionality of the embedding increases.

Experiments without context to existing literature and are not annotation free
----------------------------------------

The paper describes that existing methods are limited by their dependence on annotations. The paper does not describe existing methods using annotations designed to address tasks/applications suggested for the new model. The paper does not compare developed model to these existing methods. To make the experiments performed in the paper computationally tractable, RNA-seq reads are aligned to the reference genome (annotations of sequence) and sequences in specific regions, defined with respect to genes of interest (annotations of genes). The experiments are therefore dependent on annotation although they lose their information/interpretability in this context. The paper notes that many kmers are not included in exonic sequences according to exact matches to annotations of coding sequence. Nonetheless, these reads are in this dataset, which means they are also identified by standard tools/annotations despite their differences. In Figure 3, embedding overlap between kmers in the annotated coding sequence of ZFX and ZFY are illustrated. It would be helpful to show the proportions of reads that were used for this analysis that mapped uniquely vs not to genomic intervals in these genes (reads rather than kmers). In this regard, the absence of any note or methods describing how reads were mapped (what method, with which parameters) to reference (which reference genome, which gene annotations) is a serious limitation.

Other notes
---------------------
The heatmap in Figure 5A is ambiguous and could be improved by better annotating what is on rows/columns, perhaps also including numeric information textually in addition to by color.
Identification of kmers spanning translocation region (illustrated in embedding space by 5B) was done entirely without kmer2vec (identifying exclusive kmers, assembly of kmers, BLAST alignment to two chromosomes). Thus, claim that, as a consequence, “kmer2vec captures real genomic abnormalities and allows to extract directly from the kmer embedding space the abnormal sequence” is unsubstantiated.

---

> ### Author Response · Authors · 2018-11-27
> **Response to Reviewer 3**
>
> We would like to thank Anonymous Reviewer 3 for their thorough review.
> We hope you will reconsider your position on our paper. The paper is titled "Towards" because we are working in this direction and would like to explore this idea with the community.
>
> > Described experiments are without context to the existing literature of tools designed to address the biological challenge and by construction are not annotation free.
>
> We propose that model that would work in a reference-free fashion. In the paper in order to compute things now we filter using a reference genome in order to demonstrate how it could work. We do have several ideas to get kmer tables in a reference-free fashion, while still keeping the kmer numbers low.
>
> > See “Please stop using TopHat” (https://twitter.com/lpachter/status/937055346987712512?lang=en) by one of the authors of TopHat and the senior author of the cited paper. Standard pipelines use newer aligners like STAR which are substantially faster.
>
> We have removed from the text mention of computation speed, since it confuses the reader and our point is not about runtime, it is about the quality of the representations created by these methods
>
> > Overall, we do not believe that the described model/experiments demonstrate utility for either the specific problems in cancer genomics that motivate the paper or the biomedical research field in general.
>
> One of our goals, sorry for not stating it clearly enough, is to create a representation which would facilitate the construction of complex features to be used in an ML model which would perform supervised prediction of phenotypes or unsupervised subtyping. These goals are clearly of interest to the biomedical research community.
>
> >Experiments are for only four genes, two at a time with a 2-dimensional embedding. Unclear how patterns will hold when considering k-mers from more genes simultaneously or how embedding could scale. Model formulation suggests that k-mers from co-expressed genes will have similar embeddings, which could complicate visual inspection.
>
> During early phases of development with this model, we have performed tests with synthetic dataset on 20 randomly generated “gene” sequences. We added the plots of the kmer embedding spaces to the Appendix to answer this concern.
>
> Final note: We believe that rejection for computational complexity alone is unreasonable. The issue of scalability is a concern for many methods in our community and it does not prevent these methods from being discussed.

---

### Official Review · AnonReviewer2 · 2018-11-07
**motivation is good, the method is not well explained and is not scalable, the claims are not well supported by experiments**

**Rating:** 4
**Confidence:** 4

**Review:**

This paper aimed to dig more information into the raw RNA-seq data, in a reference-free fashion, which would not be captured and analyzed in usual RNA-seq pipelines. The method was to compute continuous embeddings for kmer sequences from the raw reads. The authors emphasized its potential use in detecting patient or tumor specific structural variations which is still a challenging task. In general, the paper is interesting. However, the novelty is limited and the developed algorithm is not yet scalable to genome-wide analysis.

- It is natural to adapt the word2vec model to biological sequencing reads, which is already seen in the literature. This paper focused on k-mer sequences, which is a good thought. But, for k-mer computing, it is important to make it fast, scalable and representative.

- "The method uses the factorized embedding model (Trofimov et al., 2017) combined with an RNN." Why does such combination work? What is your motivation to use RNN for k-mer sequences? How does it work? Unfortunately, the paper did not provide such details.

- "For all our experiments we used aligned, unquantified RNA-Seq data (BAM format files) from The Cancer Genome Atlas (TCGA) (Weinstein et al., 2013)." I was confused for this experimental settings. I thought the paper focused on reference-free fashion as mentioned from the beginning of the paper. How do you support your claims by such data collection policy?

---

> ### Author Response · Authors · 2018-11-27
> **Response to Anon Reviewer 2**
>
> We would first like to thank you for your review!
> We addressed some of the points that were made:
>
> > It is natural to adapt the word2vec model to biological sequencing reads, which is already seen in the literature. This paper focused on k-mer sequences, which is a good thought. But, for k-mer computing, it is important to make it fast, scalable and representative.
>
> While solving the scalability problem could not be done within the allowed rebuttal time period, we have added to “Limitations” section of the text some ideas we have that would make this model scalable. As for the comments from Anonymous Reviewer 2, 3 and Alex Rubinsteyn, we would like to point out that the paper is titled "Towards", because we are working in this direction and would like to explore this idea with the community. The ultimate aim of the paper was not to deliver a completely working tool, but rather to explore the possibility of using this type of model in the genomics field.
>
> > "For all our experiments we used aligned, unquantified RNA-Seq data (BAM format files) from The Cancer Genome Atlas (TCGA) (Weinstein et al., 2013)." I was confused for this experimental settings. I thought the paper focused on reference-free fashion as mentioned from the beginning of the paper. How do you support your claims by such data collection policy?
>
> To address this, we have clarified in the text this data collection shortcut in the Limitations section.
>
> > "The method uses the factorized embedding model (Trofimov et al., 2017) combined with an RNN." Why does such combination work? What is your motivation to use RNN for k-mer sequences? How does it work? Unfortunately, the paper did not provide such details.
> We added in the Method section of the text the following:
>
> “This method was previously shown to provide a multi-view embedding of genomics data. We however found that it lacked patient-specific information on the RNA sequence level. Our modified model allows for the integration of this sequence information to the factorized embeddings model.”

---

### Public Comment · ~Alex_Rubinsteyn1 · 2018-11-22
**Extremely unsatisfying evaluation**

There are already papers on reference-free methods genomics, this one is hobbled by its reliance on aligned sequences in the evaluation. If it's truly impossible to deal with the full set of kmers and filtering by alignment is always required, then it's unclear what this approach is adding.

Beyond that fundamental limitation, the scope of the evaluation is very limited and artisanal. For example, there are several mentions of detecting "patient specific mutations" in the RNA-seq reads, but only one specific translocation (PML-RARa) is considered.

Also, a quibble about the intro:

* "The transcriptome is a subset of all possible sequences of the genome that are used by the cell at any given moment and constitutes less than 2% of all genomic sequences"

I think the cited "2%" arises from a simple division of the commonly annotated human exome (~50-60mb) by the size of the full genome (~3Gb). However, the sentence sounds like it refers to the instantaneous transcript content of a single cell, and if so, it would constitute significantly less than 2% of genomic sequences. On the other hand, there is evidence of ubiquitous low-level expression of many loci in the genome which are not necessarily thought of as genes, are they considered as part of the transcriptome? Furthermore, if we consider the additional sequence diversity arising from splicing and RNA editing, then we get many sequences which are genomically templated. All that to say, I think the use of "2%" adds more confusion than clarity.

---

### Meta-Review · Area_Chair1 · 2018-12-13
**Not ready for publication**

**Confidence:** 5
**Recommendation:** Reject

**Metareview:**

This paper proposes to learn continuous of k-mer embeddings for RNA-seq analysis. Major concerns of the paper include: 1. novelty seems limited; 2. questions about the scalability of the approach; 3. evaluation experiments were not suitable for supporting the aim. Overall, this paper cannot be accepted yet.

---

> ### Author Response · Authors · 2018-12-26
> **Please make our rebuttals publicly visible.**
>
> Hello, our responses to reviewers are not visible to everyone. Can you change the visibility of the responses so that they are visible?